# Marylosides A-G, Norcycloartane Glycosides from Leaves of *Cymbidium* Great Flower ‘Marylaurencin’

**DOI:** 10.3390/molecules24132504

**Published:** 2019-07-09

**Authors:** Tatsuro Yoneyama, Kanako Iseki, Masaaki Noji, Hiroshi Imagawa, Toshihiro Hashimoto, Sachiko Kawano, Masaki Baba, Yoshiki Kashiwada, Tadahiro Yahagi, Keiichi Matsuzaki, Akemi Umeyama

**Affiliations:** 1Faculty of Pharmaceutical Sciences, Tokushima Bunri University, Yamashiro-Cho, Tokushima 770-8514, Japan; 2Kawano-Mericlone Co. Ltd., Wakimachi, Tokushima 779-3604, Japan; 3Meiji Pharmaceutical University, 2-522-1 Noshio, Kiyose, Tokyo 204-8588, Japan; 4Faculty of Pharmaceutical Sciences, Tokushima University, Tokushima 770-8505, Japan; 5School of pharmacy, Nihon University, 7-7-1 Narashinodai, Funabashi, Chiba 274-8555, Japan

**Keywords:** *Cymbidium* Great Flower ‘Marylaurencin’, Orchidaceae, maryloside, nitric oxide production, norcycloartane glycoside

## Abstract

Seven novel norcycloartane glycosides, maryloside A–G (**1**–**7**), were isolated from the leaves of *Cymbidium* Great Flower ‘Marylaurencin’, along with a known norcycloartane glycoside, cymbidoside (**8**). These structures were determined on the basis of mainly NMR experiments as well as chemical degradation and X-ray crystallographic analysis. The isolated compounds (**1**–**6** and **8**) were evaluated for the inhibitory activity on lipopolysaccharide (LPS) and interferon-γ (IFN-γ)-stimulated nitric oxide (NO) production in RAW 264.7 cells. Consequently, **1** and **3** exhibited moderate activity.

## 1. Introduction

*Cymbidium* Great Flower ‘Marylaurencin’ is a new breed produced by Kawano-Mericlone Corporation by mating. The seed is registered at the Ministry of Agriculture, Forestry and Fisheries of Japan. During our search for bioactive substances from *C*. Great Flower ‘Marylaurencin’, which is a cultivar, we have already reported some phenanthrene derivatives with antibacterial effect and cytotoxic activity, and some aromatic glycosides with antioxidant activity from its root, stem and flower [1,2,3,4]. The leaves of *C.* Great Flower ‘Marylaurencin’ are sold as an herb tea in this area. Therefore, we started exploring the active ingredients from the leaves of this plant to support peoples’ health. Here, we report the isolation and structural determination of seven novel 30-norcycloartane-type triterpene glycosides (**1**–**7**) along with one known compound, cymbidoside (**8**) [5], from the leaves of *C*. Great Flower ‘Marylaurencin’. Their structures were elucidated using high resolution (HR)-FAB-MS analyses, extensive two dimensional (2D)-heteronuclear NMR data interpretation and X-ray crystallographic analysis.

Nitric oxide (NO) plays important role as biophylaxis. On the other hand, it is involved in the overproduction of nitric oxide generated aggravation of inflammation and carcinogenesis. Nitric oxide synthase has three isoforms, eNOS (endothelial nitric oxide), nNOS (neuronal nitric oxide) and iNOS (inducible nitric oxide). Among them, especially activation of iNOS induced by such an inflammatory reaction is responsible for the overproduction of nitric oxide and contributes to DNA damage that leads to mutagenesis and carcinogenesis. We evaluated the anti-inflammatory activity of isolated compounds from *C*. Great Flower ‘Marylaurencin’ on the LPS and IFN-γ-induced NO production in mouse monocyte macrophage cell line, RAW264.7.

## 2. Results and Discussion

The leaves of *C*. Great Flower ‘Marylaurencin’ were extracted with 70% EtOH aq. at room temperature. The 70% EtOH aq. extract was partitioned into EtOAc-, BuOH- and H_2_O- soluble layers. The BuOH soluble layer was subjected to silica gel column chromatography and successively purified by reversed-phase high performance liquid chromatography (HPLC) to yield maryloside A–G (**1**–**7**) along with a known compound (**8**). 

1D and 2D NMR data of these compounds (**1**–**7**) suggested triterpene saponin. Acid hydrolysis of these compounds (**1**–**7**) afforded D-glucose, whose absolute configuration was confirmed by the method using reversed-phase HPLC developed by Tanaka [6]. 

The molecular formula of maryloside A (**1**) was determined to be C_36_H_60_O_10_ from the molecular ion peak [M + Na]^+^ at *m/z* 675.4082 (calculated for *m/z* 675.4084) in the positive-ion HR-FAB-MS with 7 degrees of unsaturation. The IR spectrum showed the presence of carbonyl group (1705 cm^−1^) and hydroxyl group (3271 cm^−1^). The ^1^H-NMR and ^13^C-NMR spectra of **1** suggested the presence of sugar moiety in the molecule (Table 1 and Table 2). The hydrolysis of **1** by cellulase liberated an aglycone of **1** (**1a**). The ^1^H-NMR spectrum of **1a** indicated characteristic signals of cyclopropane ring at δ_H_ 0.33 (doublet, d, *J* = 4.1 Hz) and 0.56 (d, *J* = 4.1 Hz), the presence of two methyl groups—signals at δ_H_ 0.86 (singlet, s) and 1.04 (s), three methyl groups at δ_H_ 1.21 (d, *J* = 6.0 Hz), 1.22 (d, *J* = 7.0 Hz) and 1.25 (d, *J* = 7.0 Hz), two oxygen-bearing methylene protons at δ_H_ 3.96 (d, *J* = 11.0 Hz) and 4.39 (d, *J* = 11.0 Hz), 4.05 (d, *J* = 11.9 Hz), 4.06 (d, *J* = 11.9 Hz) and one oxygen-bearing methine proton at δ_H_ 4.61 (d, *J* = 9.9 Hz) (Table 3). All thirty carbons in ^13^C-NMR spectrum of **1a** were clearly assigned on the basis of H-H COSY, HMQC and HMBC experiments. The ^13^C-NMR spectrum of **1a** showed the presence of a carbonyl group at δ_C_ 211.9, a tertiary alcohol at δ_C_ 76.1, a secondary alcohol at δ_C_ 69.3 and two primary alcohols at δ_C_ 57.7 and 66.8 (Table 2). The H-H COSY and HMQC spectra showed that **1a** had partial structures (**a**–**f**) –CH_2_–CH_2_– (**a**; C-1 and C-2), –CH–CH_2_– (**b**; C-4 and C-28), –CH–CH_2_–CH_2_–CH– (**c**; from C-5 to C-8), –CH_2_–CH_2_– (**d**; C-11 and C-12), –CH_2_–CH_2_–CH–CH(CH_3_)–CH–CH_2_– (**e**; from C-15 to C-23) and CH_3_–CH–CH_3_ (**f**; from C-26 to C-25 and from C-25 to C-27) (Figure 1). The HMBC spectrum showed correlations from H-1, H-2, H-5 and H-28 to C-3, and from H-19 to C-1, C-5, C-8, C-9, C-10 and C-11. These correlations established the connection between the partial structures **a** and **d**. Furthermore, the HMBC spectrum exhibited cross-peaks from H-18 to C-12, C-13, C-14 and C-17, and from H-29 to C-8, C-13, C-14 and C-15, which connected among partial structures **c**, **d** and **e**. In the HMBC spectrum, there were cross-peaks appearing from H-23 to C-24 and C-30, from H-25 to C-24 and C-30, and from H-26 and H-27 to C-24. Finally, all partial structures of **a**–**f** were connected. The relative configuration of **1a** was determined by the NOESY experiment, coupling constant and X-ray crystallographic analysis. The NOESY spectrum between H-4β-axial and H-19, between H-8β-axial [δ_H_ 1.58 (dd, *J* = 4.9, 12.6 Hz)] and H-18, and between H-8β-axial and H-19 confirmed to be β-axial oriented protons at C-4, C-8 C-18 and C-19. On the other hand, the NOESY spectrum between H-1α-axial and H-5α-axial, between H-6α-equatorial and H-28, between H-7α-axial and H-29, between H-12α-axial and H-29, and between H-17α-axial and H-29 confirmed to be α-axial-oriented protons at C-5, C-17 and C-29 positions. The configuration at C-20 position of **1a** was suggested as *S** configuration according to NOE correlations between H-12β -equatorial and H-21, and between H-18 and H-20. Finally, 20*S**, 22*R** and 24*R** were confirmed by X-ray crystallographic analysis (Figure 2). 

Sugar moiety of **1** was estimated as one D-glucose from acid hydrolysis with the method of Tanaka using reversed-phase HPLC [6] and chemical shift of ^13^C-NMR spectra. **1** was hydrolyzed by 0.5 M HCl. After drying in vacuo, the residue was dissolved in pyridine containing L-cysteine methyl ester hydrochloride and heated at 60 °C for 1 hr. The reaction mixture was directly analyzed by reversed-phase HPLC. The peak at 10.626 min was coincided with the derivative of D-glucose (10.607 min), whereas that of L-glucose was 10.142 min. The HMBC correlation of **1** from anomeric proton at H-1’ to C-28 indicated that glucose was connected to C-28 (Figure 3). The coupling constant of anomeric proton at H-1’ as *J* = 7.8 Hz revealed to be β-D-glucopyranosyl linkage. From the above data, structure of **1** was elucidated as 24-methyl-30-norcycloart-22*R**, 24*R**, 28, 30-tetraol-3-one-28-*O*-β-D-glucopyranoside (Figure 4).

The HR-FAB-MS of maryloside B (**2**) showed a molecular ion peak at *m/z* 853.4570 [M + Na]^+^, corresponding to the molecular formula C_42_H_70_O_16_. The IR spectrum of **2** showed the presence of carbonyl group (1697 cm^−1^) and hydroxyl group (3356 cm^−1^). The ^1^H and ^13^C-NMR spectra of **2** were very similar to those of **8** except for sugar moiety (Table 1). The ^1^H-NMR spectrum of **2** showed two anomeric proton doublets at δ_H_ 4.97 (H-1’) and δ_H_ 5.37 (H-1″) in the downfield region. The β-anomeric configuration of the D-glucose moieties were determined from their coupling constants of H-1’ (*J* = 7.7 Hz) and H-1″ (*J* = 7.7 Hz), respectively. Two glucose moieties (S_1_ in Figure 4) were determined to be connected to C-28 position by HMBC correlations from H-1″ to C-2’ and from H-1’ to C-28. The hydroxyl group at C-25 (δ_C_ 75.4) was determined by HMBC correlations from H-26 and H-27 to C-25. The X-ray crystallographic analysis of aglycone of known compound **8** was measured to determine the relative configuration of **2** because aglycone of both compounds have the same planner structure. The ^1^H and ^13^C-NMR values of aglycone of **2** were good agreement with those of aglycone of **8**. Consequently, the structure of **2** was elucidated to be 24-methyl-30-norcycloart-22*R**, 24*S**, 25, 28, 30-pentaol-3-one-28-*O*-β-D-glucopyranosyl (1→2)-β-D-glucopyranoside (Figure 4).

The HR-FAB-MS analysis of maryloside C (**3**) displayed a pseudomolecular ion peak at *m/z* 813.4664 [M − H]^-^, indicating the molecular formula C_42_H_70_O_15_. The IR spectrum showed the presence of carbonyl group (1699 cm^−1^) and hydroxyl group (3340 cm^−1^). The ^1^H-NMR and ^13^C-NMR spectra of **3** were very similar to those of **1** except for the additional glucose moiety (Table 1 and Table 2). The ^1^H-NMR spectrum of **3** showed two anomeric proton doublets at δ_H_ 4.87 (*J* = 7.8 Hz) and δ_H_ 5.15 (*J* = 7.8 Hz) in the downfield region, indicating two β-linked sugar moieties. HMBC correlations from the anomeric proton of H-1’ to C-28 and from the anomeric proton of H-1″ to C’-6 indicated the presence of a disaccharide moiety (S_2_ in Figure 4) at C-28. Therefore, maryloside C (**3**) was identified as 24-methyl-30-norcycloart-22*R** 24*R**, 28, 30-tetraol-3-one-28-*O*-β-D-glucopyranosyl (1→6)-β-D-glucopyranoside (Figure 4).

The molecular formula of maryloside D (**4**), C_42_H_70_O_15_, was determined by HR-FAB-MS *m/z* 837.4648 [M + Na]^+^ and ^13^C-NMR data. The IR spectrum showed the presence of carbonyl group (1686 cm^−1^) and hydroxyl group (3393 cm^−1^). Detailed examination of 1D and 2D NMR spectra of **4** and comparison with those of **1** showed that the chemical shift values of **4** were very similar with those reported for **1**, except for the additional extra sugar moiety linked to C-30 that was confirmed by the HMBC correlation from the anomeric proton H-1‴ at δ_H_ 5.00 (d, *J* = 7.8 Hz) to C-30 (Table 1 and Table 2). Thus, the structure of maryloside D (**4**) was established as 24-methyl-30-norcycloart-22*R**, 24*R**, 28, 30-tetraol-3-one-28-*O*-β-D-glucopyranosyl-30-*O*-β-D-glucopyranoside (Figure 4).

The molecular formula, C_42_H_70_O_16_ determined by HR-FAB-MS *m/z* 829.4549 [M−H]^-^ and ^13^C-NMR data, of maryloside E (**5**) was the same as those of **2**. The IR spectrum showed the presence of carbonyl group (1703 cm^−1^) and hydroxyl group (3370 cm^−1^). Detailed examination of 1D and 2D NMR spectra and HR-FAB-MS analysis of **5** showed that the chemical shift values of **5** were almost superimposable with those reported for **2** (Table 1 and Table 2). HMBC correlations from the anomeric proton signal at H-1″ at δ_H_ 5.15 (d, *J* = 7.8 Hz) to the carbon resonance at C-6’ and from H-1’ at δ_H_ 4.88 (d, *J* = 8.0 Hz) to C-28 indicated the presence of a disaccharide moiety (S_2_ in Figure 4) at C-28. Therefore, maryloside E (**5**) was identified as 24-methyl-30-norcycloart-22*R**, 24*S**, 25, 28, 30-pentaol-3-one-28-*O*-β-D-glucopyranosyl (1→6) -β-D-glucopyranoside (Figure 4).

The negative FAB-MS of maryloside F (**6**) gave a molecular ion peak at *m/z* 753.4033 [M − H]^-^, corresponding to the molecular formula C_39_H_62_O_14_. The IR spectrum showed the presence of carbonyl group (1711 cm^−1^) and hydroxyl group (3150 cm^−1^). The ^1^H-NMR and the ^1^^3^C-NMR spectra of **6** were similar to those of **2**, but a few differences were also recognized. The ^13^C-NMR spectrum of **6** showed the presence of two carboxylate esters at δ_C_ 169.6 and 168.1. The HMBC correlation from anomeric proton H-1’ at δ_H_ 4.90 (d, *J* = 7.8 Hz) to C-28 indicated that the glucose connected to C-28. The HMBC correlations from H-6’ to C-α and from H-β to C-α and C-γ indicated that malonyl moiety connected to C-6’. From the above data, the structure of maryloside F (**6**) was elucidated as 24-methyl-30-norcycloart-22*R**, 24*S**, 25, 28, 30-pentaol-3-one-28-*O*-(6′-*O*-malonyl) β-D-glucopyranoside (Figure 4).

The molecular formula of maryloside G (**7**) was determined to be C_48_H_82_O_19_ from the molecular ion peak [M − H]^-^ at *m/z* 961.5522 (calcd. for *m/z* 961.5524). The IR spectrum showed the presence of hydroxyl group (3367 cm^−1^). The ^1^H-NMR and ^1^^3^C-NMR spectra also suggested that **7** was the cycloartane-type saponin with three sugar moieties (Table 1 and Table 2). The ^1^H-NMR spectrum of **7** showed the presence of three anomeric protons at δ_H_ 4.98 (d, *J* = 7.7 Hz), 5.40 (d, *J* = 8,0 Hz) and 5.43 (d, *J* = 9.1 Hz), two methyl groups—signals at δ_H_ 0.85 (s) and 0.99 (s), four methyl groups at δ_H_ 1.13 (d, *J* = 6.9 Hz), 1.22 (d, *J* = 6.7 Hz), 1.27 (d, *J* = 6.9 Hz) and 1.41(d, *J* = 6.5 Hz), one oxygen-bearing methylene protons at δ_H_ 4.06 (d, *J* = 12.4 Hz) and 4.26 (d, *J*=12.4 Hz), and two oxygen-bearing methine protons at δ_H_ 3.45 (ddd, *J* = 10.6, 10.6, 4.6 Hz) and 4.56 (m). The ^13^C-NMR spectrum of **7** showed no carbonyl groups. HMBC correlations from a methyl group at δ_H_ 1.41 to C-3, C-4 and C-5 showed that this methyl connected to C-4 position (Figure 5). Moreover, an oxygen-bearing methine proton at δ_H_ 3.45 revealed to be at C-3 position from 1D and 2D NMR data. From the coupling pattern of H-3 at δ_H_ 3.45 (ddd, *J* = 10.6, 10.6, 4.6 Hz), the configurations of H-3 and H-4 found to be α-axial and β-axial protons, respectively. HMBC correlations from the anomeric proton at H-1’ to C-3, from the anomeric proton at H-1″to at C-2’ and from the anomeric proton at H-1‴ to at C-24 indicated that two glucose (S_1_ in Figure 4) attached to the C-3 position and another one glucose attached C-24 position. Furthermore, from the coupling constants of these anomeric protons, three sugar moieties were identified as all β-glucose (H-1’, δ_H_ 4.98, *J* = 7.7 Hz), (H-1″, δ_H_ 5.43, *J* = 9.1 Hz) and (H-1‴, δ_H_ 5.40, *J* = 8.0 Hz). From above data, structure of maryloside G (**7**) was elucidated as to be 24-methyl-30-norcycloart-3, 22*R**, 24*R**, 30-tetraol-3-*O*-β-D-glucopyranosyl (1→2)-β-D-glucopyranosyl-24-*O*-β-D-glucopyranoside (Figure 4). 

Marylosides A–G (**1**–**7**) and cymbidoside (**8**) were evaluated for the inhibitory activity of NO production in the LPS and IFN-γ-stimulated RAW 264.7 cells. Furthermore, **1** and **3** showed the inhibitory activity in dose dependently, the IC_50_ value was calculated as 17.8 ± 2.3 and 83.9 ± 4.8 µM, respectively, while IC_50_ of AG was calculated as 81.4 ± 2.6 µM (Figure 6). Other compounds showed weak inhibitory activities (IC_50_ values were ≧100 µM). The result of MTT cell viability assay revealed that all compounds hardly affect to cell viability. Considering these results, the hydroxy group at position 25 reduced the activity. Additionally, the comparison of **1** and **4** suggests the presence of glucose moiety at position 30 which also reduced the NO producing inhibitory activity.

## 3. Materials and Methods

### 3.1. General Procedures

Melting points were determined with a Yanagimoto micro melting point apparatus (Yanaco, Kyoto, Japan). Optical rotation was performed on JASCO DIP-1000 polarimeter (JASCO, Tokyo, Japan). IR spectra were measured on a JASCO FT/IR-5300. NMR spectra were obtained on a Varian UNITY 600 NMR spectrometer (Varian, CA, USA) with deuterated solvents (MeOH-*d*_4_ and pyridine-*d*_5_), and the solvent chemical shifts were taken as the internal standard. The chemical shifts are given in δ (ppm), and coupling constants are reported in Hz. HR-FAB-MS spectra were obtained on a JEOL JMS-700 instrument (JEOL, Tokyo, Japan). Kieselgel 60 (230–400 mesh, Merck) and Sephadex LH-20 (GE Healthcare Life Sciences, IL, USA) were used for column chromatography. HPLC separation was performed on a JASCO PU 1580 pump with a JASCO UV-970 detector with a CAPCELL PAK C_18_ AQ (SHISEIDO, Tokyo, Japan, 20 mm i.d. × 250 mm).

### 3.2. Plant Material

Plant growth conditions of inside a plastic greenhouse of Kawano-Mericlone Corporation in Tokushima Prefecture, Japan. Insolation; Cymbidium was grown in a plastic greenhouse. A 50% shaded net was used in summer. The average temperature was 12–30 °C. The average humidity was 40–90%.

The powder of leaves of *C.* Great Flower ‘Marylaurencin’ were supplied by Kawano-Mericlone Corporation in Tokushima Prefecture, Japan. *C.* Great Flower ‘Marylaurencin’ (Ministry of Agriculture, Forestry and Fisheries of Japan, seed registration No. 2841) was cultivated and harvested in November 2008 at Kawano-Mericlone Co., Ltd. and were identified by one of the authors (Dr. S. Kawano). A voucher specimen (TB 5430) has been deposited in the Herbarium of Faculty of Pharmaceutical Sciences, Tokushima Bunri University, Tokushima, Japan.

### 3.3. Extraction and Isolation

The leaves were dried after harvest and made into powder and stored at −20 degrees until we started the research. The powder of leaves of *C*. Great Flower ‘Marylaurencin’ (2.0 kg) were extracted with 70% EtOH aq. at room temperature for 3 months. The 70% EtOH aq. extract was dried in vacuo and partitioned into EtOAc-, BuOH- and H_2_O-soluble layers and concentrated in vacuo. The BuOH-soluble portion (42.5 g) was subjected to flush silica gel column chromatography by gradually raising the polarity with isopropylether-MeOH-H_2_O (25:6:0.1–0:80:20) to afford six fractions. Each fraction was repeatedly subjected to silica gel column chromatography using increasing concentrations of MeOH and H_2_O in chloroform as eluent. Further purification was carried out by chromatography on Sephadex LH-20 using MeOH and by repeated gradient HPLC (MeOH/H_2_O solvent system; 60–80% MeOH for **1**, 45–75% MeOH for **2**–**8**, flow rate: 3 mL/min,; UV wave length: 205 nm to afford **1** (87.5 mg), **2** (440.5 mg), **3** (66.1 mg), **4** (34.0 mg), **5** (52.3 mg), **6** (36.1 mg), **7** (14.5 mg) and **8** (106.0 mg). Their structures were elucidated by the analysis of HR-FAB-MS and ^1^H and ^13^C-NMR spectra, including 2D NMR experiments and X-ray crystallographic analysis.

*maryloside A* (**1**): m.p. 145.5–146.5 °C.; [α${]}_{D}^{23}$ + 21.9 (*c* 1.1, MeOH); FT-IR (film) 3271, 1705, 1090 cm^−1^; ^1^H-NMR (600 MHz, CD_3_OD): See Table 1; ^13^C-NMR (150 MHz, CD_3_OD): See Table 2; HR-FAB-MS *m/z* [M + Na]^+^ 675.4082 (Calcd. for C_36_H_60_O_10_Na, 675.4084)

*maryloside B* (**2**): m.p. 151.4–152.7 °C; [α${]}_{D}^{23}$ + 12.3 (*c* 1.1, MeOH); FT-IR (film): 3356, 2922, 1697, 1090 cm^−1^; ^1^H-NMR (600 MHz, C_5_D_5_N): See Table 1; ^13^C-NMR (150 MHz, C_5_D_5_N): See Table 2; HR-FAB-MS *m/z* [M + Na]^+^ 853.4570 (calcd. for C_42_H_70_O_16_Na, 853.4561)

*maryloside C* (**3**): m.p. 135.5–136.7 °C; [α${]}_{D}^{23}$ + 11.8 (*c* 1.7, MeOH); FT-IR (film): 3340, 1699, 1094 cm^−1^; ^1^H-NMR (600 MHz, C_5_D_5_N): See Table 1; ^13^C-NMR (150 MHz, C_5_D_5_N): See Table 2; HR-FAB-MS *m/z* [M − H]^-^ 813.4664 (calcd. for C_42_H_69_O_15_, 813.4636)

*maryloside D* (**4**): m.p. 133.0–134.3 °C; [α${]}_{D}^{23}$ + 10.9 (*c* 1.3, MeOH)**;** FT-IR (film): 3393, 1686, 1086 cm^−1^; ^1^H-NMR (600 MHz, C_5_D_5_N): See Table 1; ^13^C-NMR (150 MHz, C_5_D_5_N): See Table 2; HR-FAB-MS *m/z* [M + Na]^+^ 837.4648 (calcd. for C_42_H_70_O_15_Na, 837.4613)

*maryloside E* (**5**): m.p. 123.6–125.2 °C; [α${]}_{D}^{23}$ + 6.5 (*c* 2.2, MeOH). FT-IR (film) 3370, 1703, 1078, 1042 cm^−1^; ^1^H-NMR (600 MHz, C_5_D_5_N): See Table 1; ^13^C-NMR (150 MHz, C_5_D_5_N): See Table 2; HR-FAB-MS *m/z* [M − H]^–^ 829.4549 (calcd. for C_42_H_69_O_16_, 829.4586)

*maryloside F* (**6**): m.p. 119.2–121.7 °C; [α${]}_{D}^{23}$ + 19.4 (*c* 1.9, MeOH); FT-IR (film) 3150, 1744, 1711 cm^−1^; ^1^H-NMR (600 MHz, C_5_D_5_N): See Table 1; ^13^C-NMR (150 MHz, C_5_D_5_N): See Table 2; HR-FAB-MS *m/z* [M − H]^-^ 753.4033 (calcd. for C_39_H_61_O_14_: 753.4062)

*maryloside G* (**7**): m.p. 219.2–220.8 °C; [α${]}_{D}^{23}$ + 17.9 (*c* 0.8, MeOH); FT-IR (film): 3367, 1074, 1040 cm^−1^; ^1^H-NMR (600 MHz, C_5_D_5_N): See Table 3; ^13^C-NMR (150 MHz, C_5_D_5_N): See Table 3; HR-FAB-MS: *m/z* [M−H]^-^ 961.5522 (calcd. for C_48_H_81_O_19_: 961.5524)

*cymbidoside* (**8**): [α${]}_{D}^{23}$ + 18.9 (*c* 0.5, MeOH); ^13^C-NMR (150 MHz, C_5_D_5_N): 210.2 (C-3), 105.3 (C-1’), 78.5 (C-3’), 78.5 (C-5’), 77.6 (C-24), 75.4 (C-25), 75.4 (C-2’), 71.7 (C-4′), 69.5 (C-22), 66.2 (C-30), 65.6 (C-28), 62.8 (C-6’), 56.0 (C-4), 49.6 (C-17), 48.6 (C-14), 47.6 (C-8), 45.9 (C-13,) 43.7 (C-20), 41.2 (C-2), 41.1 (C-5), 35.8 (C-15), 34.2 (C-23), 33.0 (C-12), 32.1 (C-1), 28.9 (C-10), 27.7 (C-16), 27.2 (C-11), 26.9 (C-19), 26.11 (C-26), 26.07 (C-27), 25.8 (C-6), 25.4 (C-7), 24.8 (C-9), 19.6 (C-29), 18.3 (C-18) and 12.6 (C-21)

MALDI-TOF-MS *m/z* [M + Na]^+^ 691.4028 (calcd. for C_36_H_60_O_11_Na: 691.3033)

### 3.4. Determination of Nitric Oxide (NO) Production and MTT Cell Viability Assay

The concentration of nitrite in the medium was measured as indicator of NO production according to the Griess method [7]. Cells were cultured in F-12 HAM medium with 10% fetal bovine serum, 100 U/mL penicillin G and 0.1 mg/mL streptomycin. Cells were seeded on 96-well plate at 2.4 × 10^5^ cell/well and incubated at 37 °C for 2 h. Then, medium was changed to one with LPS (final concentration: 100 ng/mL), IFN-γ (final concentration: 0.33 µg/mL) in each sample. After 24 hr of incubation at 37 °C, supernatants were collected, and NO concentrations were measured by using the Griess method. Aminoguanidine was used as positive control (final concentration: 100 µM). Cell viability was assessed by MTT assay [8]. After removal of supernatant for measurement of NO concentration, 10 µL of MTT solution (5 mg/mL in phosphate buffered saline) was added. MTT-formazan was measured by the following procedure. Data represent the mean value ± standard error of mean of experiments performed in triplicated.

### 3.5. X-ray Crystallographic Analysis

Single crystals of aglycone of maryloside A (**1****a**), obtained from MeOH solution were selected and fitted onto a glass fiber and measured at −173 °C with a Bruker Apex II ultra-diffractometer using MoKα radiation. Data correction and reduction were performed with the crystallographic package Apex II. The structure was solved by direct methods using SHELXS-97 and refined by means of full matrix least-squares based on *F*^2^ using SHELXL-97 (Sheldrick, 1997). All non-hydrogen atoms were refined anisotropically, and hydrogen atoms were positioned geometrically. A total of 325 parameters were finally considered. Final disagreement indices were R_1_ = 0.0495, wR_2_ = 0.1269 (I > 2 sigma(I)). The ORTEP plot was obtained by the program PLATON (A.L. Spek, 2009). 

Crystal data: C_30_H_50_O_5_, MW = 490, Monoclinic, space group *P2(1)*, *Z* = 2, *a* = 13.2141(15) Å, *b* = 7.1031(8)Å, *c* = 15.0890(17)Å, β = 102.2600(10)°, *V* = 1384.0(3)Å^3^, GOF = 1.031.

Single crystals of aglycone of **8** (**8a**), obtained from MeOH solution were selected and fitted onto a glass fiber and measured at −173 °C with an Apex II ultra-diffractometer (Bruker, MA, USA) using MoKα radiation by the same method as **1a**. A total parameter was finally considered. Final disagreement indices were R_1_ = 0.0527, wR_2_ = 0.1358 (I > 2 sigma(I)).

Crystal data: C_30_H_50_O_6_, MW = 506, Monoclinic, space group *P2(1)*, *Z* = 4, *a* = 11.498(2) Å, *b* = 7.3583(14)Å, *c* = 31.322(6)Å, β=93.294(3)°, *V* = 2645.6(9)Å^3^, GOF = 1.034

Crystallographic data (excluding structure factors) for the structures of **1a** and **8a** have been deposited with the Cambridge Crystallographic Data Centre as Appendix A publication numbers CCDC1901424 and CCDC1457199, respectively. Copies of the data can be obtained, free of charge, on application to CCDC, 12 Union Road, Cambridge CB2 1EZ UK (Fax: +44(0)-1223-336033 or email: deposit@ccdc.cam.ac.uk).

## 4. Conclusions

Seven novel 30-norcycloartane-type triterpene glycosides (**1**–**7**) along with one known compound (**8**) were isolated from the leaves of *C*. Great Flower ‘Marylaurencin’. Although their relative structures were estimated by using NOE correlations and *J* values, it was difficult to determined relative configuration from them regarding side chains of these compounds. The configuration at the C-20 position of these compounds was suggested as *S** configuration according to biosynthetic pathway [9,10,11]. The configuration at C-20 position of **1a** was suggested as *S** configuration according to NOE correlations between H-12β-equatorial and H-21, and between H-18 and H-20 and the relative configuration of side chain of **1** was definitely determined as 20*S**, 22*R** and 24*R** by X-ray crystallographic analysis of **1a,** eventually. The relative structure of the side chain from C-20 to C-30 of **3** was fully the same as **1.** The chemical shift values of ^13^C-NMR of **3** and **4** almost superimposable with those reported for **1**. Therefore, the relative configuration of side chain of **3** and **4** was decided the same as **1**. The same thing as **3** and **4** could be said for **2**, **5** and **6** compared with **8**. The relative configuration of **8** was determined as 20*S**, 22*R** and 24*S** by the X-ray crystallographic analysis of aglycon of **8** (Figure 7 and Figure 8). The chemical shift values of ^13^C-NMR of **2**, **5** and **6** almost superimposable with those of **8**. Therefore, configuration of side chain of **2**, **5** and **6** was decided as 20*S**, 22*R** and 24*S**.

In NO production inhibitory activity assay, **1** and **3** exhibited moderate activity whereas the result of the MTT cell viability assay revealed that all compounds hardly affect cell viability. To the best of our knowledge, this is the first report of the anti-inflammatory effect of norcycloartane glycosides while norcycloartanes are reported to show anti-bacterial activity against *Micrococcus luteus* and *Bacillus subtilis* or anti-human immunodeficiency virus activity [12,13]. Inflammation by over-production of NO was related to tissue toxicity, aging and occurrence of adenocarcinoma. Therefore, continuous drinking of tea of *C.* Great Flower ‘Marylaurencin’ may contribute to a human’s health life. Taking into consideration of all other bioactivities those we already reported about *C*. Great Flower ‘Marylaurencin’, it is worth to study *C*. Great Flower ‘Marylaurencin’ further.

## Figures and Tables

**Figure 1 molecules-24-02504-f001:**
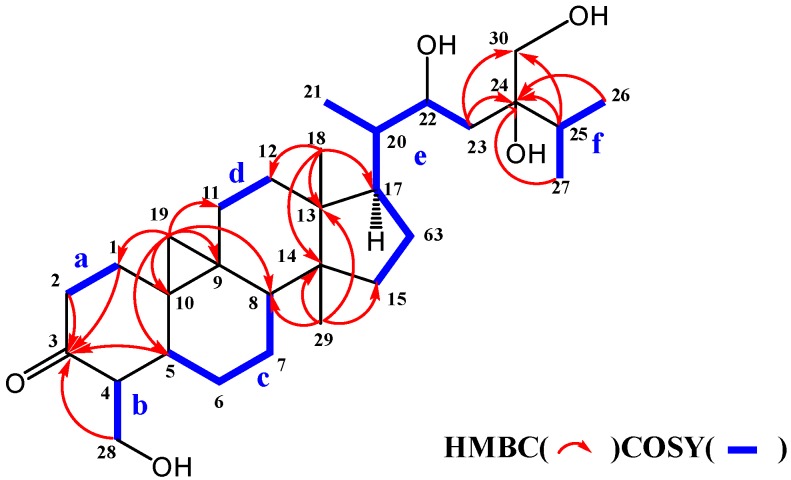
Selected COSY and HMBC correlations of **1a**.

**Figure 2 molecules-24-02504-f002:**
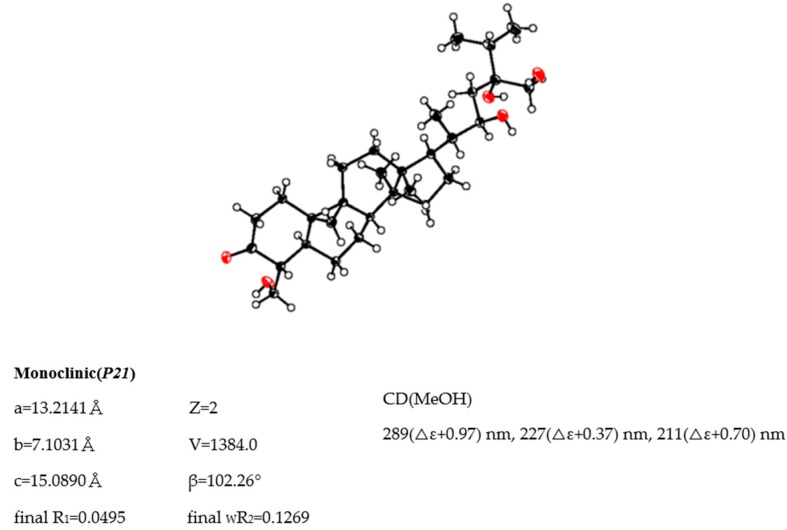
ORTEP drawing and crystal data of **1a**.

**Figure 3 molecules-24-02504-f003:**
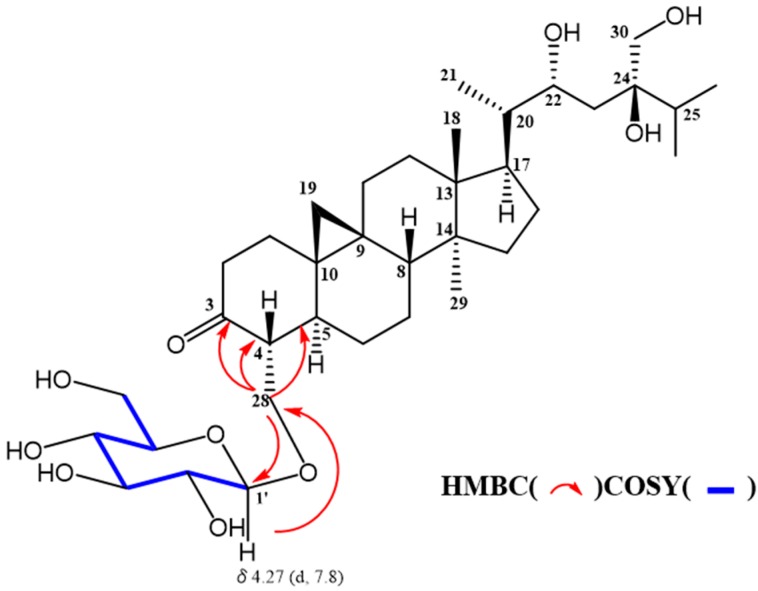
Selected COSY and HMBC data of **1**.

**Figure 4 molecules-24-02504-f004:**
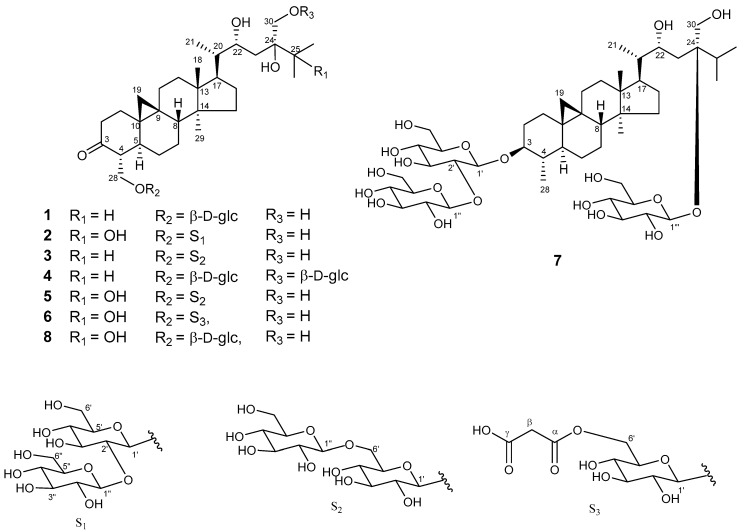
Structures of **1**–**8**.

**Figure 5 molecules-24-02504-f005:**
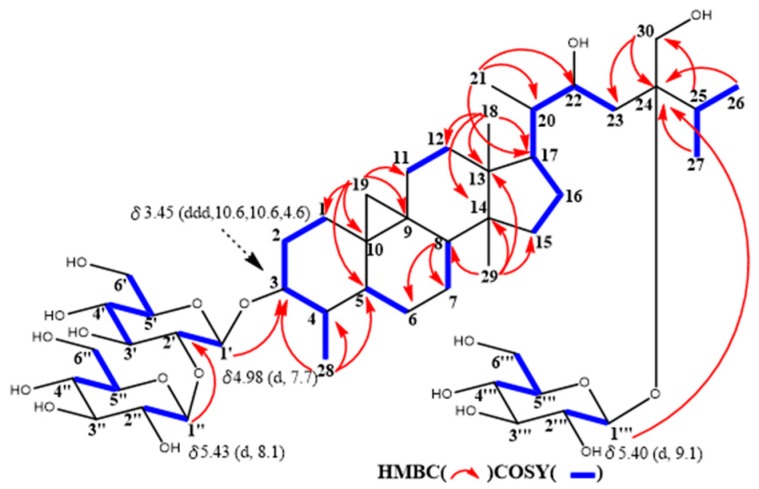
Selected COSY and HMBC correlations of **7**.

**Figure 6 molecules-24-02504-f006:**
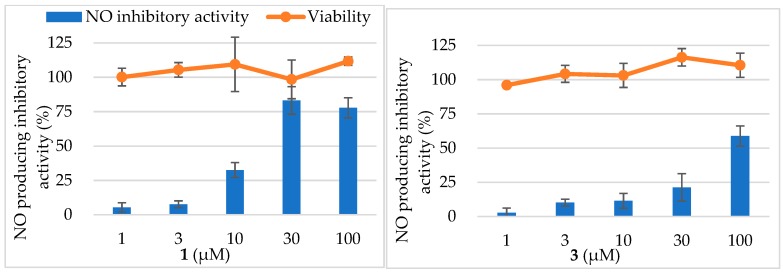
Nitric oxide (NO) production inhibitory activities and cell viability of **1** and **3**.

**Figure 7 molecules-24-02504-f007:**
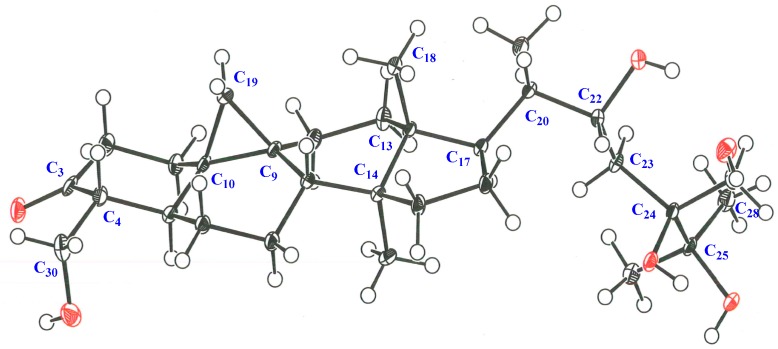
ORTEP drawing for the X-ray crystal structure of **8a**.

**Figure 8 molecules-24-02504-f008:**
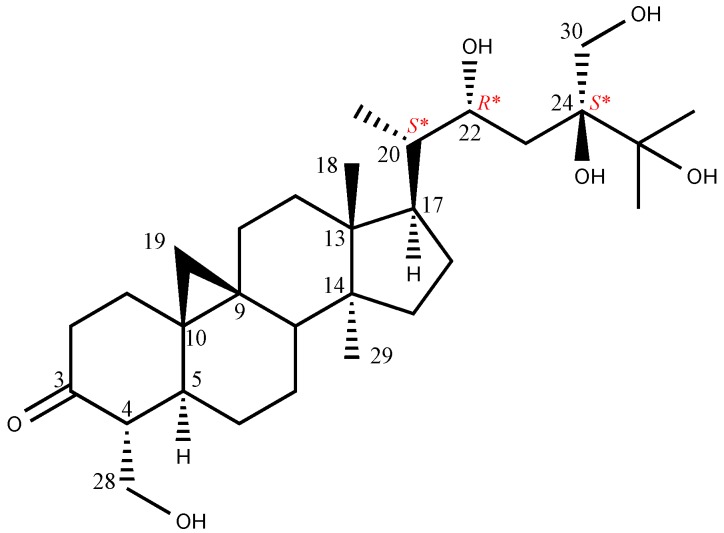
Relative configuration of **8a.**

**Table 1 molecules-24-02504-t001:** ^1^H-NMR (600 MHz) spectral data of compounds **1**–**7** [δ in ppm, coupling constants (*J* in Hz, in parenthesis)].

Position	1 ^a)^	2 ^b)^	3 ^b)^	4 ^b)^	5 ^b)^	6 ^b)^	7 ^b)^
1	1.62 (1H, m)	1.40 (1H, m)	1.37 (1H, ddd, 10.8, 7.3, 2.9)	1.38 (1H, ddd, 10.7, 4.26, 4.26)	1.36 (1H, ddd, 13.3, 6.7, 4.9)	1.38 (1H, ddd, 13.5, 7.2, 2.7)	1.44 (1H, m)
	1.90 (1H, m)	1.95 (1H, ddd, 12.9, 12.9, 4.9)	1.68 (1H, m)	1.69 (1H, m)	1.64 (1H, m)	1.69 (1H, m)	1.16 (1H, m)
2	2.41 (1H, m)	2.40 (1H, m)	2.33 (1H, m)	2.34 (1H, m)	2.33 (1H, m)	2.32 (1H, m)	1.75 (1H, m)
	2.46 (1H, m)	2.57 (1H, ddd, 11.7, 4.9, 4.9)	2.49 (1H, ddd, 11.8, 4.8, 2.9)	2.50 (1H, m)	2.48 (1H, ddd, 14.4, 4.9, 2.7)	2.48 (1H, ddd, 14.6, 4.8, 2.7)	2.54 (1H, m)
3							3.45 (1H, ddd, 10.6, 10.6, 4.6)
4	2.37 (1H, dt, 11.8, 4.2)	2.40 (1H, m)	2.25 (1H, dt, 11.8, 3.8)	2.33 (1H, m)	2.24 (1H, dt, 11.9, 3.8)	2.25 (1H, dt, 11.8, 3.2)	1.59 (1H, m)
5	2.06 (1H, ddd, 11.8, 11.8, 5.1)	2.198 (1H, m)	2.15 (1H, m)	2.09 (1H, ddd, 12.4, 12.4, 4.8)	2.11 (1H, m)	2.10 (1H, m)	1.17 (1H, m)
6	0.83 (1H, dddd, 12.4, 12.4, 12.4, 1.8)	0.75 (1H, m)	0.74 (1H, m)	0.72 (1H, m)	0.73 (1H, m)	0.71 (1H, m)	0.50 (1H, m)
	1.88 (1H, m)	1.92 (1H, m)	2.03 (1H, m)	1.95 (1H, m)	2.02 (1H, m)	2.00(1H, m)	1.63 (1H, m)
7	1.17 (1H, dddd, 12.4, 12.4, 12.4, 2.4)	1.07 (1H, m)	0.93 (1H, m)	0.93 (1H, m)	0.91 (1H, m)	0.90 (1H, m)	0.96 (1H, m)
	1.34 (1H, m)	1.21 (1H, m)	1.22 (1H, m)	1.11 (1H, m)	1.17 (1H, m)	1.14 (1H, m)	1.18 (m)
8	1.68 (1H, m)	1.57 (1H, m)	1.49 (1H, dd, 2.5, 2.5)	1.49 (1H, m)	1.48 (1H, m)	1.48 (1H, m)	1.52 (m)
11	2.13 (2H, m)	1.16 (1H, ddd, 11.1, 7.0, 7.0)	1.94 (1H, m)	1.13 (1H, m)	1.12 (1H, m)	1.12 (1H, m)	1.16 (m)
		2.05 (1H, m)	1.13 (1H, ddd, 13.3, 9.7, 4.8)	1.94 (1H, m)	1.92 (1H, m)	1.92 (1H, m)	1.89 (m)
12	1.54 (1H, m)	1.64 (2H, ddd, 11.1, 7.0, 7.0)	1.62 (2H, m)	1.61 (2H, m)	1.58 (2H, m)	1.62 (2H, m)	1.60 (2H, m)
	1.57 (1H, m)						
15	1.35 (2H, m)	1.28 (2H, m)	1.22 (2H, m)	1.25 (2H, m)	1.24 (2H, m)	1.22 (2H, m)	1.18 (2H, m)
16	1.39 (1H, m)	1.45 (1H, m)	1.47 (1H, m)	1.48 (1H, m)	1.46 (1H, m)	1.46 (1H, m)	1.37 (m)
	1.95 (1H, m)	2.32 (1H, m)	2.12 (1H, m)	2.18 (1H, m)	2.30 (1H, m)	2.0(1H, m)	2.12 (m)
17	1.63 (1H, m)	1.88 (1H, m)	1.78 (1H, d, 9.3)	1.80 (1H, dd, 10.7, 10.7)	1.85 (1H, m)	1.86 (1H, m)	1.80 (m)
18	1.09 (3H, s)	1.03 (3H, s)	1.00 (3H, s)	1.02 (3H, s)	1.00 (3H, s)	1.00 (3H, s)	0.99 (3H, s)
19	0.46 (1H, d, 4.1)	0.30 (1H, d, 4.0)	0.26 (1H, d, 4.0)	0.27 (1H, d, 4.0)	0.25 (1H, d, 3.8)	0.27 (1H, d, 4.0)	0.10 (d, 3.7)
	0.67 (1H, d, 4.1)	0.55 (1H, d, 4.0)	0.43 (1H, d, 4.0)	0.46 (1H, d, 4.0)	0.43 (1H, d, 3.8)	0.45 (1H, d, 4.0)	0.33 (d, 3.7)
20	1.68 (1H, m)	2.08 (1H, m)	2.01 (1H, m)	2.02 (1H, m)	2.06 (1H, m)	2.06 (1H, ddd, 7.0, 3.4, 3.3)	2.08 (m)
21	0.92 (3H, d, 6.5)	1.22 (3H, d, 6.6)	1.19 (3H, d, 6.9)	1.19 (3H, d, 6.6)	1.22 (3H, d, 6.7)	1.22 (3H, d, 7.0)	1.22 (3H, d, 6.7)
22	4.07 (1H, dd, 10.2, 4.2)	4.68 (1H, dd, 9.7, 2.7)	4.59 (1H, dd, 10.4, 2.3)	4.56 (1H, dd, 5.1, 2.6)	4.70 (1H, dd, 9.9, 2.9)	4.70 (1H, dd, 9.9, 3.3)	4.56 (m)
23	1.70 (1H, m)	2.197 (1H, m)	1.99 (1H, m)	1.99 (1H, m)	2.36 (1H, m)	2.15 (1H, m)	2.02 (m)
		2.34 (1H, m)	2.11 (1H, m)	1.61 (1H, m)	2.19 (1H, dd, 14.7, 9.9)	2.32 (1H, m)	2.08 (m)
25	1.93 (1H, m)		2.35 (1H, sep, 7.0)	2.46 (1H, sep, 9.2)			2.48 (sext, 6.9)
26	0.95 (3H, d, 7.1)	1.69 (3H, s)	1.25 (3H, d, 7.0)	1.18 (3H, d, 9.2)	1.694 (3H, s)	1.69(3H, s)	1.27 (3H, d, 6.9)
27	0.964 (3H, d, 7.3)	1.68 (3H, s)	1.22 (3H, d, 7.0)	1.17 (3H, d, 9.2)	1.687 (3H, s)	1.68(3H, s)	1.13 (3H, d, 6.9)
28	3.87 (1H, dd, 10.2, 2.7)	4.18 (1H, m)	4.22 (1H, dd, 10.2, 3.8)	4.19 (1H, dd, 10.4, 3.0)	4.21 (1H, m)	4.27 (1H, dd,10.3,3.2)	1.41 (3H, d, 6.5)
	4.07 (1H, dd, 10.2, 4.2)	4.42 (1H, m)	4.45 (1H, dd, 10.2, 3.8)	4.44 (1H, dd, 10.4, 4.0)	4.45 (1H, dd, 10.2, 3.8)	4.40 (1H, m)	
29	0.957 (3H, s)	0.85 (3H, s)	0.81 (3H, s)	0.83 (3H, s)	0.79 (3H, s)	0.79 (3H, s)	0.85 (3H, s)
30	3.49 (1H, d, 11.4)	4.12 (1H, d, 10.7)	4.04 (1H, d, 11.1)	4.17 (1H, d, 11.1)	4.37 (1H, d, 11.2)	4.36 (1H, d, 11.3)	4.06 (d, 12.4)
	3.52 (1H, d, 11.4)	4.39 (1H, d, 10.7)	4.06 (1H, d, 11.1)	4.49 (1H, d, 11.1)	4.41 (1H, d, 11.2)	4.40 (1H, d, 11.3)	4.26 (d, 12.4)
glucoside at C-28 or C-3							
1’	4.27 (1H, d, 7.8)	4.97 (d, 7.7)	4.87 (1H, d, 7.8)	4.97 (1H, d, 8.8)	4.88 (1H, d, 8.0)	4.90 (1H, d, 7.8)	4.98 (1H, d, 7.7)
2’	3.13 (1H, dd, 9.1, 7.8)	4.17 (1H, m)	3.97 (1H, dd, 8.8, 7.8)	4.03 (1H, dd, 8.8, 8.8)	3.97 (1H, dd, 8.8, 8.0)	4.00 (1H, dd, 8.8, 7.8)	4.26 (1H, m)
3’	3.35 (1H, dd, 9.1, 9.1)	4.209 (1H, m)	4.17 (1H, dd, 9.0, 8.8)	4.24 (1H, m)	4.17 (1H, dd, 8.8, 8.8)	4.19 (1H, dd, 8.8, 8.8)	4.34 (dd, 11.5, 11.5)
4’	3.28 (1H, dd, 9.6, 9.6)	4.207 (1H, m)	4.12 (1H, dd, 9.0)	4.25 (1H, m)	4.12 (1H, dd, 9.1, 8.8)	4.08 (1H, dd, 9.1, 8.8)	4.21 (dd, 11.5, 11.5)
5’	3.25 (1H, ddd, 9.6, 9.6, 2.3)	3.94 (1H, m)	4.06 (1H, m)	3.94 (1H, ddd, 5.5, 2.7, 2.7)	4.06 (1H, m)	4.05 (1H, m)	3.89 (ddd, 11.5, 5.4, 2.6)
6’	3.66 (1H, dd, 11.8, 9.6)	4.40 (1H, dd, 11.7, 5.2)	4.38 (1H, dd, 11.5, 5.2)	4.38 (1H, dd, 11.4, 5.5)	4.35 (1H, dd, 11.5, 6.0)	4.89 (1H, dd, 11.6, 5.9)	4.35 (dd, 11.5, 11.5)
	3.85 (1H, dd, 11.8, 2.3)	4.55 (1H, dd, 11.7, 2.7)	4.83 (1H, dd, 11.5, 1.7)	4.54 (1H, dd, 11.4, 2.7)	4.83 (1H, dd, 11.5, 1.7)	5.06 (1H, dd, 11.6, 1.6)	4.49 (dd, 11.5, 3.0)
1’’		5.37 (1H, d, 7.7)	5.15 (1H, d, 7.8)		5.15 (1H, d, 7.8)		5.43 (d, 9.1)
2’’		4.04 (1H, dd, 9.1, 7.7)	4.05 (1H, m)		4.05 (1H, dd, 7.8, 6.5)		4.15 (dd, 9.1, 9.1)
3’’		4.28 (1H, dd, 9.1, 9.1)	4.24 (1H, m)		4.23 (1H, dd, 6.5, 6.5)		4.24(m)
4’’		4.197 (1H, m)	4.24 (1H, m)		4.24 (1H, dd, 6.5, 6.0)		4.28 (dd, 9.5, 1.9)
5’’		3.86 (1H, ddd, 7.7, 5.4, 2.6)	3.94 (1H, ddd, 8.8, 6.3, 3.4)		3.94 (1H, ddd, 3.2, 6.0, 8.9)		3.94(ddd,9.5,4.7,3.3)
6’’		4.37 (1H, m)	4.35 (1H, dd, 11.8, 6.3)		4.36 (1H, m)		4.41 (dd, 11.9, 4.7)
		4.49 (1H, dd, 11.8, 2.6)	4.52 (1H, dd, 11.8, 3.4)		4.53 (1H, dd, 11.7, 3.2)		4.52 (dd, 11.9, 3.3)
malonyl moiety β						3.78 (2H, d, 4.8)	
glucoside at C-28 or C-3							
1’’’				5.00 (d, 7.8)			5.40 (d, 8.0)
2’’’				4.07 (dd, 7.8, 4.1)			4.12 (dd, 9.1, 8.0)
3’’’				4.20 (m)			4.26 (dd, 9.1, 3.1)
4’’’				4.19 (m)			4.13 (m)
5’’’				3.96 (ddd, 9.2, 6.3, 3.3)			4.02 (ddd, 9.7, 6.7, 3.0)
6’’’				4.38 (dd, 11.4, 6.3)			4.25 (m)
				4.54 (dd, 11.4, 3.3)			4.54 (m)

^a)^ Spectra recorded in CD_3_OD. ^b^^)^ Spectra recorded in C_5_D_5_N.

**Table 2 molecules-24-02504-t002:** ^13^C-NMR (150 MHz) spectral data of compounds **1**–**7**, **1a**, and **8a** (δ in ppm).

Position	1 ^a)^	1a ^b)^	2 ^b)^	3 ^b)^	4 ^b)^	5 ^b)^	6 ^b)^	7 ^b)^	8a ^a)^
1	33.1	32.1	32.2	31.9	32.0	31.9	31.8	30.9	33.0
2	41.7	41.3	41.2	41.0	41.1	41.0	41.0	34.0	41.6
3	213.8	211.9	210.8	210.1	210.2	210.1	210.1	86.5	213.8
4	56.7	57.6	56.2	55.9	56.0	55.9	55.9	43.9	58.1
5	42.1	40.4	41.3	40.8	41.2	40.8	40.8	43.8	41.8
6	26.6	25.8	26.0	25.8	25.8	25.8	25.7	24.9	26.5
7	26.3	25.3	25.5	25.3	25.4	25.3	25.3	25.3	26.1
8	48.6	47.4	47.7	47.5	47.5	47.5	47.5	46.8	48.6
9	26.1	24.7	24.8	24.7	24.8	24.7	24.7	23.6	25.9
10	30.0	28.8	28.9	28.8	28.9	28.8	28.8	30.0	29.7
11	28.1	27.1	27.2	27.1	27.1	26.8	27.1	27.1	28.0
12	34.1	33.0	33.0	33.0	33.0	33.0	33.0	33.1	34.2
13	46.8	45.9	45.9	45.9	45.9	45.8	45.8	45.9	46.7
14	49.6	48.6	48.6	48.5	48.6	48.6	48.6	48.7	49.5
15	36.8	35.8	35.8	35.8	35.9	35.8	35.8	35.6	36.6
16	28.4	27.7	27.7	27.8	27.8	27.7	27.7	27.6	28.3
17	50.2	49.4	49.6	49.4	49.4	49.6	49.6	49.4	50.2
18	18.5	18.1	18.4	18.2	18.2	18.2	18.3	17.9	18.6
19	27.7	26.8	27.2	26.8	26.8	27.1	26.8	26.9	27.5
20	44.2	43.6	43.8	43.6	43.9	43.9	43.9	43.6	44.3
21	12.5	12.6	12.6	12.6	12.7	12.6	12.6	12.7	12.5
22	70.3	69.3	69.5	69.4	69.3	69.4	69.4	68.1	70.2
23	33.9	34.1	34.2	34.1	33.2	34.2	34.2	32.2	33.8
24	76.9	76.1	77.6	76.1	76.2	77.6	77.6	83.4	77.9
25	36.1	35.7	75.4	35.7	34.9	75.4	75.4	33.9	76.2
26	17.9	18.0	26.1	18.0	17.9	26.1	26.1	18.1	25.6
27	17.4	17.6	26.1	17.6	17.7	26.1	26.1	17.7	25.6
28	65.9	57.7	66.2	66.9	65.6	65.4	65.4	15.2	58.1
29	19.8	19.4	19.6	19.5	19.5	19.5	19.5	19.3	19.8
30	66.8	66.8	65.7	65.4	75.9	66.2	66.2	65.3	66.0
glucoside at C-28 or C-3									
1’	104.9		103.3	105.1	105.3	105.1	105.1	104.1	
2’	75.1		82.7	75.2	75.4	75.2	75.2	83.6	
3’	77.9		78.1	78.3	78.5	78.3	78.3	78.3	
4’	71.6		71.9	71.7	71.6	71.7	71.5	71.5	
5’	78.0		78.5	77.4	78.6	77.4	75.2	78.0	
6’	62.8		63.1	70.1	63.0	70.1	65.5	63.0	
1’’			105.6	105.5		105.5		106.0	
2’’			76.6	75.3		75.3		76.9	
3’’			78.3	78.4		78.4		78.0	
4’’			71.4	71.7		71.7		71.8	
5’’			78.5	78.6		78.5		78.4	
6’’			62.6	62.8		62.8		62.9	
malonyl moiety									
α							168.1		
β							42.8		
γ							169.6		
glucoside at C-30 or C-24									
C-1‴					106.1			97.6	
C-2‴					75.4			75.8	
C-3‴					78.5			79.2	
C-4‴					71.6			72.2	
C-5‴					78.6			78.4	
C-6‴					63.0			63.0	

^a)^ Spectra recorded in CD_3_OD. ^b^^)^ Spectra recorded in C_5_D_5_N.

**Table 3 molecules-24-02504-t003:** ^1^H-NMR (600 MHz) spectral data of compounds **1a** and **8a** [δ in ppm, coupling constants (*J* in Hz, in parenthesis)].

Position	1a ^b)^	8a ^a)^
1	1.45 (1H, ddd, 13.2, 6.9, 2.8)	1.53 (1H, ddd, 13.5, 6.6, 3.3)
	1.83 (1H, m)	1.76 (1H, m)
2	2.41 (1H, ddd, 14.1, 6.9, 6.9)	1.98 (1H, ddd, 11.9, 11.9, 6.6)
	2.53 (1H, ddd, 14.1, 2.8, 2.8)	2.37 (1H, m)
4	2.251 (1H, m)	2.18 (1H, ddd, 11.9, 4.7, 2.6)
5	2.245 (1H, m)	2.37 (1H, m)
6	0.76 (1H, m)	0.75 (1H, dddd, 12.9, 12.9, 12.9, 2.7)
	1.95 (1H, m)	1.72 (1H, m)
7	1.07 (1H, dddd, 12.6, 12.6, 12.6, 2.9)	1.08 (1H, dddd, 12.9, 12.9, 12.9, 2.7)
	1.25 (1H, m)	1.72 (1H, m)
8	1.58 (1H, dd, 12.6, 4.9)	1.61 (1H, m)
11	2.02 (1H, m)	1.22 (2H, m)
	1.19 (1H, ddd, 14.0, 9.7, 4.8)	
12	1.65 (2H, m)	1.81 (2H, m)
15	1.27 (2H, m)	1.30 (2H, m)
16	1.49 (1H, m)	1.35 (1H, m)
	2.15 (1H, m)	2.04 (1H, m)
17	1.80 (1H, dd, 11.0, 9.1)	1.65 (1H, m)
18	1.04 (3H, s)	1.01 (3H, s)
19	0.33 (1H, d, 4.1)	0.38 (1H, d, 4.1)
	0.56 (1H, d, 4.1)	0.58 (1H, d, 4.1)
20	2.05 (1H, m)	1.72 (1H, m)
21	1.21 (3H, d, 6.0)	0.93 (3H, d, 6.6)
22	4.61 (1H, d, 9.9)	4.14 (1H, dd, 9.7, 2.7)
23	2.12 (1H, d, 10.6)	1.62 (1H, m)
	2.00 (1H, m)	1.85 (1H, d, 14.8)
25	2.37 (1H, sep, 7.0)	
26	1.22 (3H, d, 7.0)	1.29 (3H, s)
27	1.25 (3H, d, 7.0)	1.27 (3H, s)
28	3.96 (1H, d, 11.0)	3.66 (1H, dd, 11.3, 4.7)
	4.39 (1H, d, 11.0)	3.98 (1H, dd, 11.3, 2.6)
29	0.86 (3H, s)	0.87 (3H, s)
30	4.05 (1H, d, 11.9)	3.82 (1H, d, 11.3)
	4.06 (1H, d, 11.9)	3.97 (1H, d, 11.3)

^a)^ Spectra recorded in CD_3_OD. ^b)^ Spectra recorded in C_5_D_5_N.

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
