# Peer review of "Marylosides A-G, Norcycloartane Glycosides from Leaves of Cymbidium Great Flower ‘Marylaurencin’"

_molecules, 2019, doi:10.3390/molecules24132504_

Round 1
Reviewer 1 Report
Authors have included requested biochemical data for 2 compounds analyzed at different concentrations. It justifies their postulated inhibitory effects on nitric oxide formation.
Data included in this revised manuscript support the postulated inhibitory effects of the identified compounds on formation of NO.
Revision by a Native speaker is recommended.
Minor remarks
Fig 6 legend – use the term nitric oxide or the chemical formula NO instead of ‘No’
Author Response
Thank you very much for your proper and kindly review, and for your time.
Revision by a Native speaker is recommended.
I corrected some misspells in the paper.
Table1 and Table2 maronyl → malonyl
Page 7 336 Phenathrene → Phenanthrene
Page 7 349 Pharm.Bull. → Pharm. Bull.
Page 7 354 macropage → macrophage
Minor remarks
Fig 6 legend – use the term nitric oxide or the chemical formula NO instead of ‘No’
I used the chemical formula NO instead of ‘No’.
Reviewer 2 Report
The manuscript “Marylosides A-G, norcycloartane glycosides from 2 leaves of Cymbidium Great Flower ‘Marylaurencin’” [molecules-553204-peer-review-v1] submitted by Kanako Iseki, Sakiko Tani, Tatsuro Yoneyama, Masaaki Noji, Kazuko Yoshikawa, Hiroshi Imagawa, Toshihiro Hashimoto, Sachiko Kawano, Masaki Baba, Yoshiki Kashiwada and Akemi Umeyama has been revised by the authors. The reviewer is grateful to the authors for taking to heart the previous comments and for addressing several concerns.
All comments of the reviewer have been addressed and the equality and readability of the manuscript have been improved. The manuscript should hence be acceptable for publication in Molecules
Author Response
Our paper has been improved thanks to you.
Thank you very much for everything (your proper and kindly review), and for your time.
This manuscript is a resubmission of an earlier submission. The following is a list of the peer review reports and author responses from that submission.
Round 1
Reviewer 1 Report
This manuscript describes the structures of seven novel glycosides of norcycloartane type, isolated by the authors from the leaves of Cymbidium Great Flower ‘Marylaurencin’. Their structures have been elucidated using NMR, HR MS and X-ray crystallographic analysis. Preliminary data on the test of their biological activity is provided too.
Comments
Information on the systematic localization of the plant used in this study has to be mentioned in the Introduction. Why the paper Yang et al., BMC Evol Biol. 2013 describing identification of Cymbidium species is not commented here ? It definitely deserves it.
Biological activity test has to be described in the broader context. Several important information is missing:
· comment on what is already known on the biological activity of structurally related compounds
· brief comment on the methodology of biological tests already applied in the literature
· why the particular cell line (RAW 264.7) has been selected
· why aminoguanidin was used as a positive control
· why the only parameter evaluated was % of inhibition of NO production – brief comment has to be transferred from Conclusions (l. 303-312). and placed in the section Results
· why there is no data on the toxicity of the analyzed compounds
· why the particular concentration (100 μM) was applied
· biological effect of the selected 3 compounds has to be tested at least for 5 different concentrations
As it is the biological data are too preliminary, moreover the novelty of the presented results is not clear.
Description of the experimental protocol is not sufficient.
· general procedures – details of the conditions of measurements are not provided. For NMR – conditions of measurement, solvent used; for HPLC – was the UV detector used ? at which wavelength ?
· plant growth conditions (e.g. insolation, average temperature, average humidity at the certain geographical location) should be provided
· please describe in details the protocol of fractionation of the extract (l.194): what do you mean saying ‘70 % aq EtOH extract was partitioned into EtOAc-, BuOH- and H2O-soluble portions’ ? I guess the ethanolic extract was evaporated and subsequently the residue was mixed with EtOAc. The solution was separated and a residue was subsequently dissolved in BuOH. Again the solution was separated and a residue was finally dissolved in water. Is it correct?
· provide the details of separation, e.g. for flush chromatography (l.198) - what was the gradient of MeOH and H2O in chloroform; similarly for HPLC (l.199)
· neither the RAW 264.7 cell line growth conditions nor the method of lipopolysaccharide stimulation and MTT assay are described.
Minor remarks
l.35 – the name of the organism tested in LPS treatment has to be mentioned in t he Abstract
l.37 – replace ‘70 % aq EtOH’ with ‘ 70 % EtOH aq’
l.39 – replace ‘portions’ with ‘fractions’
l.46 – replace ‘7 degrees of unsaturation’ with ‘seven unsaturated C=c bonds’
l.51 and below – please change the style here and through the entire manuscript accordingly:
· ‘the presence of two singlet methyls at δH 0.86 (s) and 1.04 (s)’ replace with: ‘the presence of two methyl groups – signals at δH 0.86 (singlet, s) and 1.04 (s)’
· ‘three doublet methyls at’ with ‘three methyl groups at… (doublet, d)’
l.189 – it seems the leaves were stored for 10 years – what were the conditions of storage? are the analyzed compound stable upon these conditions?
l.288 – a reference is missing in the sentence ‘..according to biosynthetic pathway’
l.303 – nitric oxide is not a radical – please correct
Author Response
Please find my Word file below.

Reviewer 2 Report
The paper describes norcycloartane glycosides from leaves of cymbidium great flower ‘Marylaurencin’. Seven novel norcycloartane glycosides were isolated. Their structures were determined on the basis of NMR experiments as well as chemical degradation and X-ray crystallographic analysis. The paper could be published subject to revisions as indicated below:
1. Page 2, lines 78-79
“The coupling constant of anomeric proton at H-1' as J=7.8 revealed to be β-D-glucopyranosyl linkage”
Some of the well-established references in the international literature should be provided, i.e. Carbohydrates & Glycoconjugates, Encyclopedia of Nuclear Magnetic Resonance, 2002, p.p.1107-1134.
2. Page 10, lines 193-194
“The powder of leaves of C. Great Flower ‘Marylaurencin’ (2.0 kg) were extracted with 70 % aq EtOH at room temperature.”
The time duration of the extraction processes should be provided.
2. Page 13, lines 311-312.
“… hardly affect to cell damage.”
The English should be improved.
4. The number of references is very limited and most of them are from their own group.
5. Supplementary Material
In all Supplementary Figures, the assignment of the most prominent peaks and connectivities should be provided.
Furthermore, in each figure a caption, explaining the NMR experimental parameters, should be provided.
Author Response
Please find my Word file below.

Reviewer 3 Report
The manuscript “Marylosides A-G, norcycloartane glycosides from 2 leaves of Cymbidium Great Flower ‘Marylaurencin’” [molecules-324159-peer-review-v1] submitted by Kanako Iseki, Sakiko Tani, Tatsuro Yoneyama, Masaaki Noji, Kazuko Yoshikawa, Hiroshi Imagawa, Toshihiro Hashimoto, Sachiko Kawano, Masaki Baba, Yoshiki Kashiwada and Akemi Umeyama describes isolation and structural characterization of seven new and one know maryloside glucosides from Cymbidium Great Flower ‘Marylaurencin’. Furthermore inhibition of NO production in LPS stimulated RAW cells has been studied.
In particular 1D and 2D NMR and MS as well as some chemical analyses and X-ray analysis of one derivative have been used to determine the structures. All investigations (including inhibition of NO production) are performed with modern and common state of the art analytical methods. The overall work seems quite well planned and performed. The structure determination has been made quite carefully with respect to entire NMR data and especially to known cymbidoside. The reported results sound perspicuous and are well described. The reviewer does find only very minor incongruity in the argumentation about the structures.
The results possess some importance in furthering our knowledge of natural products from Cymbidium Great Flower ‘Marylaurencin’ and glycosylation pattern on norcycloartanes. The manuscript is hence of some interest in the fields of Phytochemistry, NMR Spectroscopy, Organic Chemistry, and Carbohydrate Chemistry. However, it is not yet completely in a form to be published in “Molecules”. Therefore, there are some comments, which should be taken into account by the authors prior to acceptance of the manuscript:
1) The authors are encouraged to give an explanation in the manuscript, why they have studied inhibition of NO production in LPS stimulated RAW cells cause by these natural products. Are there any reasons to make this particular biochemical/immunological test with these compounds?
2) The authors should briefly point out their technique to identify the D-glucose for all samples (line 76-77). On one hand site the should give a few details and results from the Tanaka-method. In addition they should briefly refer to the NMR-data of the glucosides to identify the structure of the glycoside.
3) Structure determination of compound 8 should be briefly given before compound 2 is described. In this context NMR data of compound 8 are mentioned.
4) Using negative or positive mode in HR-FAB-MS as well as calculated masses should always be given for all compounds in results and discussion.
5) The authors should give a brief explanation why the use C5D5N as solvent for the NMR measurements. NMR chemical shifts are difficult to compare with similar stuctures, nomaly measured in CD3OD. Have the compounds 2-7 been badly soluble in methanol?
6) Chapter 3.3: The table style of analytical data is nice, but uncommon, The authors should use a commons style of presentation.
7) The are some minor inaccuracies in English language. (e.g.. line 27: “a” instead of “an”, etc). Probably a native speaker can correct such (very minor) mistakes.
Author Response
Please find my Word file below.

Round 2
Reviewer 1 Report
I found the revised manuscript partially improved. Below please find my previous comments which hve not ignored by the authors. In my opinion respective comments have to appear in the body of the manuscript.
- Information on the systematic localization of the Cymbidium species has to be mentioned in the Introduction.
- Biological activity test has to be described in the broader context - biological effect of the selected 3 compounds has to be tested at least for 5 different concentrations
As it is the biological data are too preliminary, moreover the novelty of the presented results is limited.
- Description of the experimental protocol is not sufficient.
· plant growth conditions (e.g. insolation, average temperature, average humidity at the certain geographical location) should be provided – these data are critical for future analyses of the content and profile of analyzed metabolies
· details of separation, e.g. for flush and HPLC chromataography – did you use the isocratic elution?
Minor remarks
l.60 and below – authors use jargon when describing the NMR spectra – the style has to be changed through the entire manuscript – please keep in mind that methyls cannot be either singlet or doublet – instead they are observed as singlet or doublet signals in the spectrum, see below for examples:
· ‘the presence of two singlet methyls at δH 0.86 (s) and 1.04 (s)’ replace with: ‘the presence of two methyl groups – signals at δH 0.86 (singlet, s) and 1.04 (s)’
· ‘three doublet methyls at’ with ‘three methyl groups at… (doublet, d)’
l.205 – it seems the leaves were stored for 10 years – what were the conditions of storage? are the analyzed compound stable upon these conditions? – this section has to be modified – how long the powder or extracts were stored ? Otherwise the stability of the analyzed compounds has to be discussed
Author Response
Dear Reviewer 1 Thank you for your precise review. The answers to Reviewer's questions & corrections are in the attached document. I think I could correc most of them, but it's very difficult for me about the biological activity and I would like you to give the best advice. Thank you. Sincerely yours, Akemi Umeyama
